# Detection of malignant peripheral nerve sheath tumors in patients with neurofibromatosis using aneuploidy and mutation identification in plasma

Austin K Mattox[1†], Christopher Douville[1†], Natalie Silliman[1], Janine Ptak[1], Lisa Dobbyn[1], Joy Schaefer[1], Maria Popoli[1], Cherie Blair[1], Kathy Judge[1], Kai Pollard[2,3], Christine Pratilas[2], Jaishri Blakeley[4], Fausto Rodriguez[5], Nickolas Papadopoulos[1,3,5], Allan Belzberg[6*], Chetan Bettegowda[1,3,6*]

[1]Ludwig Center for Cancer Genetics and Therapeutics, Sidney Kimmel Comprehensive Cancer Center, Johns Hopkins University School of Medicine, Baltimore, United States; [2]Department of Pediatrics, Johns Hopkins University School of Medicine, Baltimore, United States; [3]Department of Oncology, Sidney Kimmel Comprehensive Cancer, Johns Hopkins University, Baltimore, United States; [4]Department of Neurology, Johns Hopkins University School of Medicine, MD School of Medicine, Baltimore, United States; [5]Department of Pathology, Johns Hopkins University School of Medicine, Baltimore, United States; [6]Department of Neurosurgery, Johns Hopkins University School of Medicine, Baltimore, United States

*For correspondence:
abelzbe1@jhmi.edu (AB);
cbetteg1@jhmi.edu (CB)

[†]These authors contributed equally to this work

**Abstract** Malignant peripheral nerve sheath tumors (MPNST) are the deadliest cancer that arises in individuals diagnosed with neurofibromatosis and account for nearly 5% of the 15,000 soft tissue sarcomas diagnosed in the United States each year. Comprised of neoplastic Schwann cells, primary risk factors for developing MPNST include existing plexiform neurofibromas (PN), prior radiotherapy treatment, and expansive germline mutations involving the entire *NF1* gene and surrounding genes. PN develop in nearly 30–50% of patients with neurofibromatosis type 1 (NF1) and most often grow rapidly in the first decade of life. One of the most important aspects of clinical care for NF1 patients is monitoring PN for signs of malignant transformation to MPNST that occurs in 10–15% of patients. We perform aneuploidy analysis on ctDNA from 883 ostensibly healthy individuals and 28 patients with neurofibromas, including 7 patients with benign neurofibroma, 9 patients with PN and 12 patients with MPNST. Overall sensitivity for detecting MPNST using genome wide aneuploidy scoring was 33%, and analysis of sub-chromosomal copy number alterations (CNAs) improved sensitivity to 50% while retaining a high specificity of 97%. In addition, we performed mutation analysis on plasma cfDNA for a subset of patients and identified mutations in *NF1*, *NF2*, *RB1*, *TP53BP1*, and *GOLGA2*. Given the high throughput and relatively low sequencing coverage required by our assay, liquid biopsy represents a promising technology to identify incipient MPNST.

## Editor's evaluation

The manuscript explores the use of liquid biopsy to detect MPNST, which is a rare malignant peripheral nerve sheath tumor. The current manuscript presents clinical relevant data of interest to the field and should be published without further delay.

## Introduction

Neurofibromatosis type 1 (NF1) is caused by inherited or de novo mutations in the *NF1* gene that codes for the cytoplasmic protein neurofibromin (*DeClue et al., 1992*). Neurofibromin is a GTPase-activating protein (GAP) for the RAS family of proto-oncogenes, and mutations in *NF1* lead to persistent RAS signaling and uncontrolled cellular growth through downstream RAF, MEK, and ERK signaling (*DeClue et al., 1992*; *Carroll, 2012*). Activated RAS resulting from the loss of GTPase activity of NF1 also leads to downstream activation of the PI3K/AKT/mTOR pathway, further contributing to increased proliferation (*Carroll, 2012*).

Malignant peripheral nerve sheath tumors (MPNST) are the deadliest cancer that arises in individuals diagnosed with NF1 and account for nearly 5% of the 15,000 soft tissue sarcomas diagnosed in the United States each year (*Rasmussen et al., 2001*). Comprised of neoplastic Schwann cells, primary risk factors for developing MPNST include existing plexiform neurofibromas (PN), prior radiotherapy treatment, and expansive germline mutations involving the entire *NF1* gene and surrounding genes (*De Raedt et al., 2003*). PN develop in nearly 30–50% of patients with NF1 and most often grow rapidly in the first decade of life. One of the most important aspects of clinical care for NF1 patients is monitoring PN for signs of malignant transformation to MPNST that occurs in 10–15% of patients.

Bi-allelic loss of *NF1* is not sufficient for malignant transformation of PN to MPNST (*Zheng et al., 2008*; *Yang et al., 2008*; *Zhu et al., 2002*). Additional mutations or copy number alterations of genes such as *TP53*, *SUZ12*, *EGFR*, *CDKN2A*, and *TERT* that are often not present in benign PN suggest that these alterations represent advanced progression to atypical neurofibroma (AN) and MPNST (*Cichowski et al., 1999*; *De Raedt et al., 2014*; *Legius et al., 1994*; *Perry et al., 2002*; *Zhang et al., 2014*).

Data from National Cancer Institute (NCI) NF1 Natural History Study suggest that nearly 50% of patients with PN develop well-demarcated nodular areas within the PN that are larger than 3 cm in size, lack the central dot sign characteristic of PN, and typically show more rapid growth (*Mautner et al., 2008*). These distinct nodular lesions (DNL) correlate with pain, and biopsy/resection of DNL leads to a confirmed diagnosis of AN in 70% of cases (*Higham et al., 2016*). Of all confirmed AN cases in the study, all were DNL by MRI and were associated with a modest FDG update of [SUV] = 2.7.

Despite these preliminary results, MRI is unable to reliably differentiate between benign and malignant tumors (*Derlin et al., 2013*). Additional studies have suggested FDG-PET has sensitivities of nearly 90% in symptomatic patients, but only when using an SUV cutoff of 3.5 and a non-standard clinical protocol of delayed imaging at 4 hr (*Ferner et al., 2008*). Using similar criteria, FDG-PET may have similar sensitivities for monitoring asymptomatic patients for malignant transformation, but only at 49.5% specificity (*Azizi et al., 2018*). Pathologically, there are no standardized pathognomonic genetic alterations or immunohistochemical stains to differentiate MPNST from other sarcomas. While gross specimens that clearly arise from nerves lend credence to a diagnosis of MPNST, negative staining for cytokeratins and melanoma markers like Melan-A, MITF, and HMB45 can be useful in distinguishing MPNST from carcinoma and melanoma (*Reinert et al., 2019*; *Pekmezci et al., 2015*; *Fletcher, 2014*). S100 expression is also decreased or completely lost in MPNST (*Pekmezci et al., 2015*). Genomic loss of *NF1* and *CDKN2A* are thought to be lost early in disease progression but testing for these mutations is also not clinically standardized.

Given the lack of specific imaging and pathologic diagnostic criteria to diagnose MPNST, more accurate and cost-effective biomarkers are needed. Liquid biopsies that assay for mutations or aneuploidy in circulating tumor DNA (ctDNA) represent an attractive, minimally invasive option that could be performed at each longitudinal patient visit. Mutations in polycomb repressive complex 2 (PRC2) subunits such as *SUZ12* and *EED* are found in nearly 70% of MPNST (*De Raedt et al., 2014*; *Zhang et al., 2014*). Mutations in β-III-spectrin have also been found in up to 90% of MPNST (*Hirbe et al., 2018*). Additional Ras pathway activating mutations in genes such as *PIK3CA*, *KIT*, *PDGFRA*, *PTPN11*, *FGFR1*, and RASSF9, and cell-cycle gene mutations in genes such as *RB1* and *CHEK2* have also been described (*Brohl et al., 2017*).

Liquid biopsies also have the advantage over traditional biopsies of capturing tumor heterogeneity. This is important because within a single tumor, there may be areas of PN, AN, low grade MPNST, and high grade MPNST, and traditional single-site biopsy may not capture the most malignant site.

In the present study, we perform aneuploidy analysis on 883 ostensibly healthy individuals and 28 patients with neurofibromas, including 7 patients with benign neurofibroma, 9 patients with PN, and 12 patients with MPNST. While overall sensitivity for detecting NF using genome wide aneuploidy measurements was limited, analysis of sub-chromosomal changes may be promising for detecting MPNST. In addition, we performed mutation analysis on plasma cfDNA for a subset of patients and identified mutations in *NF1*, *NF2*, *RB1*, *TP53BP2*, and *GOLGA2*.

## Results

### Patient characteristics

The primary objective of this pilot study was to differentiate MPNST from PN using genome wide and focal aneuploidy analysis of cfDNA isolated from plasma. To quantify the rate of genome wide copy number alterations (CNAs) detected in plasma cfDNA of healthy persons, we analyzed 883 samples from a previously published study (*Douville et al., 2020*) using a revised RealSeqS algorithm and a median of 10,223,275 UIDs per sample (*Supplementary file 1*). Our patient cohort included 28 patients with NF, including 7 patients with neurofibromas, 9 patients with PN, and 12 patients with MPNST, analyzed in the same manner as healthy controls with a median of 11,240,762 UIDs per sample (*Supplementary file 2*). All samples had matched leukocyte analysis to exclude germline CNAs. For patients with biopsy-confirmed MPNST, 58% (7/12) had positive PET scans, 17% (2/12) had prior chemotherapy, 17% (2/12) had prior chemotherapy and radiation, 8% (1/12) had prior radiation only, 17% (2/12) had prior surgery only, and 8% (1/12) had prior surgery, chemotherapy, and radiation. Median length of follow up was 523 days.

### Analysis of genome wide aneuploidy

Ninety-six percent (27/28) of patients enrolled onto our study met the criteria for NF1 diagnosis. RealSeqS, which amplifies approximately 750,000 loci across 39 chromosome arms, was used to calculate a genome wide aneuploidy score (GAS) to call plasma samples positive or negative at 97% specificity as determined by the 883 healthy controls. The inclusion of the large number of healthy controls is especially important because it allows for a realistic estimation of specificity and a comparison between healthy persons, patients with neurofibromas, as those with MPNST, as would be done in a real-world setting. The median GAS score in healthy controls was 0.008 ± 0.102, and 0.005 ± 0.249 and 0.018 ± 0.652 in benign/plexiform neurofibromas and MPNST, respectively (*Figure 1*). At 97% specificity in healthy controls, at the time of blood draw, the false positive rate among benign/plexiform neurofibromas was 6.3% (1/16, p = 0.42 compared to healthy controls), while the sensitivity for detecting MPNST was 33% (4/12, p < 0.001 compared to healthy controls). GAS score did not correlate with tumor volume ($R^2$ = 0.09), history of prior adjuvant therapy (p = 0.64) or PET positivity (p = 0.25), but patients who were alive at the time of last follow up had lower GAS scores (p = 0.045).

Interestingly, the one patient with the plexiform neurofibroma deemed to be a false positive at the time of blood draw (INDIA 1283, *Supplementary file 3*) had a GAS score of 0.997. Biopsy at the time of blood draw confirmed diffuse and atypical changes in the PN. Upon later review, this patient progressed to MPNST 25 months after blood draw, suggesting that aneuploidy analysis significantly pre-dated clinical progression.

### Analysis of focal copy number alterations

In addition to assessing genome wide aneuploidy, RealSeqS can detect focal amplifications and deletions across 39 chromosome arms. We profiled sub-chromosomal changes across 13 chromosome arms commonly altered in MPNST (*Figure 2*), including 4q (*PDGFRA*), 5p (*TERT*), 6q (*TBX1*), 7p (*EGFR*), 7q (*BRAF*), 8q (*EXT1*), 9p (*CDKN2A* and *CDKN2B*), 10q (*PTEN*), 11p (*EXT2*), 11q (*EED*), 15q (*IDH2*), 17p (*TP53*), and 17q (*NF1* and *SUZ12*) (*Supplementary file 4*). In benign/plexiform NF, only one patient had a focal deletion across all loci assayed. Interestingly, this patient (INDIA 1283) had a deletion in *TERT* and was the same patient that had a GAS score of 0.997 that later progressed to MPNST. Among the 12 patients with MPNST, 17% (2/12) had losses in *TERT*, 8% (1/12) had a loss at *TP53*, and 50% (6/12) had losses on 17q at *SUZ12* (*Figure 2—figure supplement 1*). These data suggest that focal changes may be useful biomarkers of progression to MPNST.

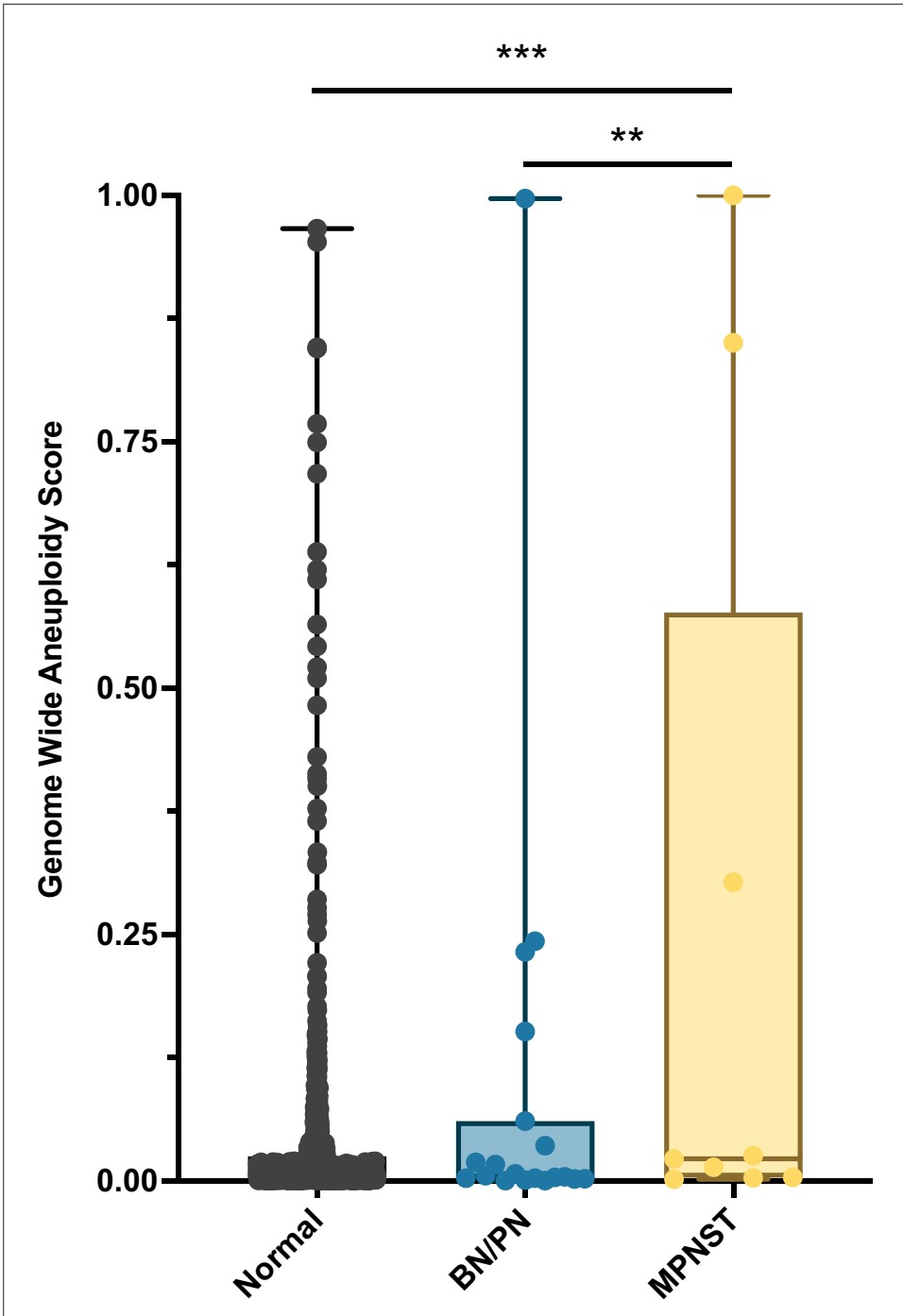

**Figure 1.** Distribution of genome wide aneuploidy (GAS) scores in healthy individuals and patients with benign (BN) or plexiform neurofibronas (PN) or MPNST. **p < 0.01, ***p < 0.001.

## ctDNA mutation analysis

Enough banked plasma was available from six patients, two with benign neurofibromas and four with MPNST, to assay ctDNA for mutations using ddPCR (***Supplementary files 1 and 5***). Three patients had positive ctDNA results, including one patient (INDIA 1280) with a benign neurofibroma of the right femoral nerve that had a *GOLGA2* splice site acceptor mutation at a mutant allele frequency (MAF) of 0.19%, and two MPNST patients (INDIA 1244 and INDIA 1281) that a *RB1* R787Qfs*23 mutation at 2.55% and a *TP53BP2* A324V mutation at 0.04%, respectively.

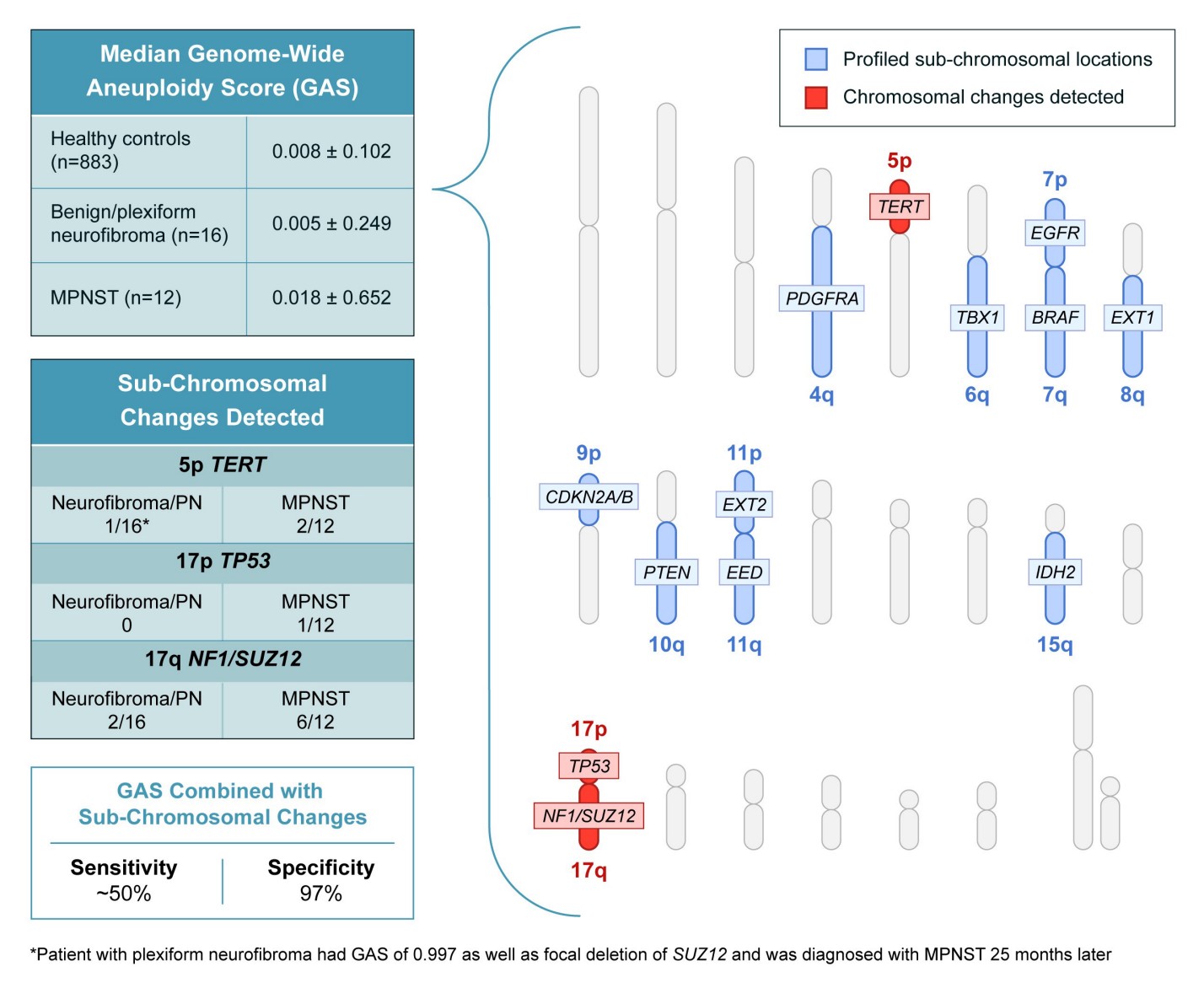

**Figure 2.** Genome wide aneuploidy scores and detection of sub-chromosomal copy number changes in *PDGFA, TERT, TBX1, EGFR, BRAF, EXT1, CDKN2A/B, PTEN, EXT2, EED, IDH2, TP53, NF1,* and *SUZ12* allow for detection of 50% of MPNST at 97% specificity.

The online version of this article includes the following figure supplement(s) for figure 2:

**Figure supplement 1.** Patients with MPNST demonstrated loss of *SUZ12* on chromosome 17q.

## Discussion

One of the major clinical challenges in caring for individuals with NF1 is to be able to identify an incipient MPNST. Currently, only anatomic and PET imaging has been shown to be effective (*Ferner and Gutmann, 2002*; *Akshintala et al., 2020*; *Canavese and Krajbich, 2011*). Use of PET imaging to track recurrence additionally requires that the primary benign neurofibroma is PET positive before treatment and necessitates the use of radionuclides (*Ferner and Gutmann, 2002*; *Reilly et al., 2017*). Thus, alternative methods to monitor disease progression to MPNST, like liquid biopsy, are greatly needed.

To our knowledge, this is the first published study to differentiate MPNST from pre-malignant neurofibromas using PCR amplicon-based aneuploidy and mutation analysis of ctDNA. During final preparation of our manuscript, *Szymanski et al., 2021* published results using different but orthogonal approaches to ours — fragment size analysis and ultra-low-pass whole genome sequencing — that

demonstrated promise for differentiating MPNST from benign precursor lesions. While our study suggests that genome wide aneuploidy scoring alone may not have high sensitivity for detecting the progression from PN to MPNST, the combination of GAS and detection of sub-chromosomal changes in genes such as *TERT*, *TP53*, and *SUZ12* or mutations in ctDNA may lead to a sensitivity of ~50% at a high specificity of 97%. Our sensitivity is likely impacted by low genome wide sequencing coverage, which may be improved by a focused panel that covers the most common CNAs in MPNST (*Zhang et al., 2014*; *Brohl et al., 2017*).

Malignant transformation of a plexiform neurofibroma to MPSNT can occur over years, and given the mean follow up is 17 months, we cannot definitively identify all cases that may have progressed. In our study, the one patient with PN who had an overwhelmingly positive GAS score of 0.997 and a focal deletion of *SUZ12* at the time of initial blood draw was diagnosed with an MPNST 25 months later.

Main limitations of our study include the relatively small number of patients, the lack of ctDNA data for all patients, and lack of follow up blood draws at fixed intervals. A key strength of our study is the relatively low sequencing coverage needed to detect both genome wide and sub-chromosomal CNAs. The ability to multiplex samples lends high throughput, as each sample only requires ~10 M reads to identify relevant CNAs. This is important for real world implementation as most aneuploidy studies typically utilize 6–10 x the amount of sequencing, greatly increasing the cost and limiting feasibility. While our data allow us to make some inferences about CNAs and disease progression from PN to MPNST, it will be important for future prospective studies to collect additional blood samples over a longer period to determine whether GAS, focal CNAs, or ctDNA positivity predicts progression and overall survival.

## Materials and methods

### Patients

All individuals participating in the study provided written informed consent after approval by the institutional review board at The Johns Hopkins IRB00075499. The study complied with the Health Insurance Portability and Accountability Act and the Deceleration of Helsinki.

### Library construction and whole exome sequencing buffer and PCR conditions

Tumor and matched lymphocytic normal DNA library preparation was performed as previously described (*Bettegowda et al., 2013*). Genomic DNA from tumor and normal samples were fragmented and used for Illumina TruSeq library construction (Illumina, San Diego, CA) according to the manufacturer's instructions. DNA was purified using Agencourt AMPure XP beads (Beckman Coulter, IN) in a ratio of 1.0–0.9 of PCR product to beads. Purified, fragmented DNA was mixed with 36 µl of $H_2O$, 10 µl of End Repair Reaction Buffer, 5 µl of End Repair Enzyme Mix (cat# E6050, NEB, Ipswich, MA). The 100 µl end-repair mixture was incubated at 20°C for 30 min, and purified using Agencourt AMPure XP beads (Beckman Coulter, IN) in a ratio of 1.0–1.25 of PCR product to beads. 42 µl of end-repaired DNA was mixed with 5 µl of 10 X dA Tailing Reaction Buffer and 3 µl of Klenow (exo-)(cat# E6053, NEB, Ipswich, MA). The 50 µl mixture was incubated at 37°C for 30 min and purified using Agencourt AMPure XP beads (Beckman Coulter, IN) in a ratio of 1.0–1.0 of PCR product to beads. 25 µl of A-tailed DNA was mixed with 6.7 µl of $H_2O$, 3.3 µl of PE-adaptor (Illumina), 10 µl of 5 X Ligation buffer and 5 µl of Quick T4 DNA ligase (cat# E6056, NEB, Ipswich, MA). The ligation mixture was incubated at 20°C for 15 min and purified using Agencourt AMPure XP beads (Beckman Coulter, IN) in a ratio of 1.0–0.95 and 1.0 of PCR product to beads.

To obtain an amplified library, twelve PCRs of 25 µl each were set up, each including 15.5 µl of $H_2O$, 5 µl of 5 x Phusion HF buffer, 0.5 µl of a dNTP mix containing 10 mM of each dNTP, 1.25 µl of DMSO, 0.25 µl of Illumina PE primer #1, 0.25 µl of Illumina PE primer #2, 0.25 µl of Hotstart Phusion polymerase, and 2 µl of the DNA. The PCR program used was: 98°C for 2 min; 12 cycles of 98°C for 15 s, 65°C for 30 s, 72°C for 30 s; and 72°C for 5 min. DNA was purified using Agencourt AMPure XP beads (Beckman Coulter, IN) in a ratio of 1.0–1.0 of PCR product to beads. Exonic regions were captured in solution using the Agilent SureSelect v.4 kit (Agilent, Santa Clara, CA). The captured library was then purified with a Qiagen MinElute column purification kit and eluted in 17 µl of 70°C EB to obtain 15 µl of captured DNA library. To amplify the captured DNA library, eight 30 µL PCR reactions containing

19 µl of H₂O, 6 µl of 5 x Phusion HF buffer, 0.6 µl of 10 mM dNTP, 1.5 µl of DMSO, 0.30 µl of Illumina PE primer #1, 0.30 µl of Illumina PE primer #2, 0.30 µl of Hotstart Phusion polymerase, and 2 µl of captured exome library were set up. The PCR program used was: 98°C for 30 s; 14 cycles of 98°C for 10 s, 65°C for 30 s, 72°C for 30 s; and 72°C for 5 min. To purify PCR products, a NucleoSpin Extract II purification kit (Macherey-Nagel, PA) was used. Paired-end sequencing resulting in 100 bases from each end of the fragments was performed using Illumina HiSeq 2,500 (Illumina, San Diego, CA).

Plasma preparation: Peripheral blood was collected in K2-EDTA tubes after informed consent was obtained, and plasma was isolated as previously described (*Diehl et al., 2008*). cfDNA from each of the plasma samples was purified using a BioChain cfDNA Extraction Kit (BioChain, cat #K5011610) using the manufacturer's recommended protocol.

## Processing of next generation sequencing data

Somatic mutations were identified using VariantDx custom software for identifying mutations in matched tumor and normal samples from whole exome sequencing (WES). Prior to mutation calling, primary processing of sequence data for both tumor and normal samples were performed using Illumina CASAVA software (v1.8), including masking of adapter sequences. Sequence reads were aligned against the human reference genome (version hg19) using ELAND software. Candidate somatic mutations, consisting of point mutations, insertions, and deletions were then identified using VariantDx. In brief, an alignment filter was applied to exclude quality failed reads, unpaired reads, and poorly mapped reads in the tumor. A base quality filter was applied to limit inclusion of bases with reported phred quality score >30 for the tumor and >20 for the normal. A mutation in the tumor was identified as a candidate somatic mutation only when (i) distinct paired reads contained the mutation in the tumor; (ii) the number of distinct paired reads containing a particular mutation in the tumor was at least 10% of the total distinct read pairs; (iii) the mismatched base was not present in >1% of the reads in the matched normal sample as well as not present in a custom database of common germline variants derived from dbSNP; and (iv) the position was covered in both the tumor and normal. Mutations arising from misplaced genome alignments, including paralogous sequences, were identified and excluded by searching the reference genome.

Candidate somatic mutations were further filtered based on gene annotation to identify those occurring in protein coding regions. Functional consequences were predicted using snpEff and a custom database of CCDS, RefSeq and Ensembl annotations using the latest transcript versions available on hg19 from UCSC (https://genome.ucsc.edu/). Predictions were ordered to prefer transcripts with canonical start and stop codons and CCDS or Refseq transcripts over Ensembl when available. Finally, mutations were filtered to exclude intronic and silent changes, while retaining mutations resulting in missense mutations, nonsense mutations, frameshifts, or splice site alterations. A manual visual inspection step was used to further remove artifactual changes.

## ddPCR

Cell-free DNA was extracted using the QIAGEN circulating nucleic acid kit (Catalog# 55114). Extracted cell-free DNA was analyzed with custom designed droplet digital PrimePCR assays using the BioRad QX200 droplet digital PCR system to determine the number of wild-type and mutant genomic equivalents following the manufacturer's recommendations. A mutation was selected for each tumor based on the results of the WES results. ddPCR was then performed in triplicate on DNA derived from the plasma and ctDNA levels were quantified. These data were used to calculate the overall MAF for each somatic mutation, defined as the number of mutant counts divided by the total number of counts for a given amplicon.

## RealSeqS

RealSeqS uses a single primer pair to amplify about 750,000 loci scattered throughout the genome (*Douville et al., 2020*). After massively parallel sequencing, gains or losses of each of the 39 chromosome arms covered by the assay were determined using a bespoke statistical learning method (*Douville et al., 2018*). A support vector machine (SVM) was used to discriminate between aneuploid and euploid samples. The SVM was trained using 2,651 aneuploid samples and 1,348 euploid plasma samples. Samples were scored as positive when the genome wide aneuploidy score was >0.28. Code is available at https://zenodo.org/record/3656943#.YaZZCdDMKUk.32 (*Douville, 2020*).

## Statistical analysis methods

Comparison of GAS at 97% specificity was conducted with a one-way ANOVA with post-hoc Tukey's correction. Clinicopathological data were compared using a (add symbol) test or linear regression with Spearman's correlation. A $p \leq 0.05$ was considered significant.

## Acknowledgements

This work was supported by grants from the NIH: 1R21CA208723-01, R37 CA230400, U01 CA230691. In addition a grant from the DOD W81XWH-16–0078, grant 2014107 from the Doris Duke Charitable Foundation and a Burroughs Wellcome Career Award for Medical Scientists all supported this work.

## Additional information

### Competing interests

Christopher Douville: is a consultant to Exact Sciences and is compensated with income and equity. Christine Pratilas: is a paid consultant for Roche/ Genentech and Day One Therapeutics; and receives research funding from Kura Oncology and Novartis Institute of Biomedical Research, all for work that is outside the scope of the submitted manuscript. Nickolas Papadopoulos: is a founder of Thrive Earlier Detection, an Exact Sciences Company. Is a consultant to Thrive Earlier Detection. Holds equity in Exact Sciences. Is a founder of and holds equity in Personal Genome Diagnostics. Is a consultant to Personal Genome Diagnostics. Is a founder and holds equity in ManaTbio. Is a founder, holds equity, and serves on the board of directors of Haystack, Inc. Holds equity in and is a consultant to CAGE Pharma. Owns equity in Neophore and is a consultant to Neophore. The companies named above as well as other companies have licensed previously described technologies related to the work described in this paper from Johns Hopkins University. Is an inventor on some of these technologies. Licenses to these technologies are or will be associated with equity or royalty payments to the inventors as well as to Johns Hopkins University. The terms of all of these arrangements are being managed by Johns Hopkins University in accordance with its conflict-of-interest policies. Chetan Bettegowda: is a consultant for Depuy-Synthes, Galectin Therapeutics and Bionaut Labs. The other authors declare that no competing interests exist.

### Funding

| Funder | Grant reference number | Author |
| --- | --- | --- |
| National Institutes of Health | 1R21CA208723-01 | Chetan Bettegowda |
| National Institutes of Health | R37 CA230400 | Chetan Bettegowda |
| National Institutes of Health | U01 CA230691 | Chetan Bettegowda |
| DOD | W81XWH-16-0078 | Austin K Mattox Allan Belzberg Chetan Bettegowda |
| Doris Duke Charitable Foundation | grant 2014107 | Chetan Bettegowda |
| Burroughs Wellcome Fund | Career Award for Medical Scientists | Chetan Bettegowda |

The funders had no role in study design, data collection and interpretation, or the decision to submit the work for publication.

### Author contributions

Austin K Mattox, Conceptualization, Data curation, Formal analysis, Methodology, Validation, Visualization, Writing – original draft, Writing – review and editing; Christopher Douville, Conceptualization, Data curation, Formal analysis, Investigation, Writing – original draft, Writing – review and editing;

Natalie Silliman, Janine Ptak, Lisa Dobbyn, Joy Schaefer, Maria Popoli, Cherie Blair, Kathy Judge, Kai Pollard, Christine Pratilas, Jaishri Blakeley, Fausto Rodriguez, Allan Belzberg, Investigation; Nickolas Papadopoulos, Formal analysis, Writing – original draft, Writing – review and editing; Chetan Bettegowda, Conceptualization, Data curation, Formal analysis, Funding acquisition, Investigation, Writing – original draft, Writing – review and editing

**Author ORCIDs**
Austin K Mattox http://orcid.org/0000-0002-7567-5542
Christopher Douville http://orcid.org/0000-0002-2510-4151
Allan Belzberg http://orcid.org/0000-0003-1158-2117
Chetan Bettegowda http://orcid.org/0000-0001-9991-7123

**Ethics**
Human subjects: All individuals participating in the study provided written informed consent after approval by the institutional review board at The Johns Hopkins IRB00075499. The study complied with the Health Insurance Portability and Accountability Act and the Deceleration of Helsinki.

**Decision letter and Author response**
Decision letter https://doi.org/10.7554/eLife.74238.sa1

---

## Additional files

**Supplementary files**
• Supplementary file 1. Genome wide aneuploidy (GAS) scores and number of useable UIDs for 883 healthy controls.
• Supplementary file 2. Clinicopathological characteristics of the patient cohort and the results of genome wide aneuploidy analysis and mutation analysis of ctDNA.
• Supplementary file 3. Detailed genome wide aneuploidy scores per chromosome arm.
• Supplementary file 4. Focal copy number change analysis for 13 commonly altered loci in MPNST.
• Supplementary file 5. Whole exome sequencing results for 6 patients with ctDNA mutation analysis.
• Transparent reporting form

**Data availability**
The code and data used in the manuscript are deposited at https://zenodo.org/record/3656943#.YaZZCdDMKUk. Non-commercial academic use is granted full use of both the scripts and data files stored in the repository. Access is granted immediately once the request has been received through the Zenodo portal. Commercial use remains restricted per Johns Hopkins Medicine legal requirements.

The following dataset was generated:

| Author(s) | Year | Dataset title | Dataset URL | Database and Identifier |
|---|---|---|---|---|
| Douville C, Karchin R, Kinzler KW, Papadopoulos N, Vogelstein B | 2020 | Software for Assessing aneuploidy with repetitive element sequencing | https://doi.org/10.5281/zenodo.3656942 | Zenodo, 10.5281/zenodo.3656942 |

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
