## [Editor Report]

The manuscript explores the use of liquid biopsy to detect MPNST, which is a rare malignant peripheral nerve sheath tumor. The current manuscript presents clinical relevant data of interest to the field and should be published without further delay.

---

## [Decision Letter]

**Decision letter after peer review:**

Thank you for submitting your article “Detection of malignant peripheral nerve sheath tumors in patients with neurofibromatosis type I using aneuploidy and mutation identification in plasma” for consideration by *eLife*. Your article has been reviewed by Wafik El-Deiry and two external experts.

The authors worked to develop a test to detect a deadly rare tumor, malignant peripheral nerve sheath tumor (MPNST), that can arise from plexiform neurofibromas that occur in 30-50% of patients with neurofibromatosis. The transformation requires more that biallelic NF1 loss and involves other gene mutations or copy number alterations in various genes including TP53, CDKN2A, EGFR, SUZ12 and TERT, among others, which are usually not in the plexiform neurofibromas. The authors wanted to improve upon clinical features, FDG-PET and MRI in detecting MPNST. They developed a liquid biopsy ctDNA test and evaluated it in 883 healthy individuals and 28 patients with neurofibromas including 12 patients with MPNST. Use of genome-wide aneuploidy scoring detected about one third of MPNST and sensitivity was improved to 50% by adding copy number alterations, in addition to mutated genes typically found in MPNST. The work addresses a clinical challenge and the progress made by the authors is clinically relevant in going beyond the limitations of currently available methods. A limitation of the current study is the number of patients with neurofibromas and MPNST who were tested and the retrospective analysis.

Please note: You have neglected to cite a recent publication that is quite similar. Here is a link to the other publication: https://journals.plos.org/plosmedicine/article?id=10.1371/journal.pmed.1003734

You can note from the abstracts that the sample sizes, methods, and results are quite similar in both cases. The NF and MPNST community is a rather small group, and the published data was presented as a plenary at the NF Conference in June.